# Reprogramming Large Pretrained Language Models for Antibody Sequence Infilling

## Abstract

Antibodies comprise the most versatile class of binding molecules, with numerous applications in biomedicine. Therapeutic antibody development requires designing novel and diverse sequences with improved properties, while maintaining the structural consistency. Computational design of antibodies involves unusual challenges relative to designing other classes of proteins, as antibodies comprise multiple long, variable, and unstructured loops at the complementarity-determining region (CDR) that determine the antigen binding affinity and specificity of an antibody. Recently, deep language models and graph neural networks have shown impressive success in antibody sequence generation. Since only a limited number of antibody structures are known, training a model using this limited data can lead to degraded performance, particularly lacking diversity in the generated samples. To address such issues, we leverage the method of Model Reprogramming (MR) here, which focuses on repurposing pretrained machine learning models for target domain tasks with scarce data, where it may be difficult to train a high-performing model from scratch. Prior works in MR have primarily focused on classification-based tasks. We extend the capabilities of reprogramming beyond classification tasks, and towards a more complex problem of antibody sequence generation. Specifically, we introduce Reprogramming for Protein Sequence Infilling, a framework in which pretrained natural language models are repurposed for protein sequence infilling via reprogramming, to infill protein sequence templates as a method of novel protein generation. For variable CDR sequence design, we formulate the task as text infilling that uses the constant region of an antibody as the sequence template. Results on antibody design benchmarks show that our reprogrammed model on low resourced antibody sequence dataset provides highly diverse CDR sequences, up to more than a two-fold increase of diversity over the baselines, without losing structural integrity and naturalness. The performance benefit of the reprogrammed model learning only from antibody sequences is more evident for longer CDR design or for multiple loop infilling at once, compared to existing graph-based models that require additional structural information. The generated sequences also demonstrate enhanced antigen binding specificity or virus neutralization ability.

## 1 Introduction

Antibodies have emerged as essential therapeutic agents in the treatment of cancer and various other autoimmune, infectious and metabolic diseases. Since 1985, approximately 100 monoclonal antibodies (mAbs) have been designated as drugs by FDA (Jin et al., 2022). Compared to small molecule drugs, the advantage of using antibody proteins as therapeutics is their high specificity resulting in less adverse effects. A key challenge in antibody design is tailoring their binding specificity, which is mainly influenced by the complementarity determining region (CDR). CDR plays a crucial role in antigen recognition and binding processes. It is composed of six hypervariable loops, three formed by each of heavy (H) and light (L) chains. Together, the CDRs shape the antigen binding site of the antibody.

Five of the six loops usually adopt well-characterized canonical conformations. In contrast, the CDR-H3 loop shows substantial variability in sequence and structure, and hence cannot be described by a canonical structure model. Even when compared to other protein loop structures, the CDR-H3 clearly stands out with its significantly higher structural diversity.

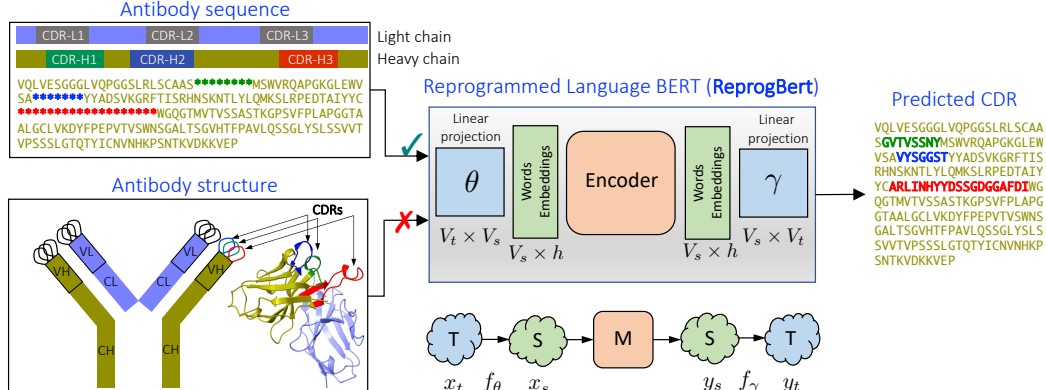

Figure 1: Overview of the proposed Protein Sequence Infilling using Model Reprogramming . Given a heavy chain of an antibody, the goal is to design three Complementarity-Determining Regions (CDR-H1, CDR-H2, CDR-H3), shown in green, blue and red colors, using information from the rest of the protein. The infilling problem is formulated similar to the masked-language modeling task, where the missing amino acids are marked with a token ⟨MASK⟩ and the model generates tokens to infill them. We emphasize that *our system is a sequence-only method*, and while the structure information might be available (bottom of the figure, showing Y-shaped antibody structure with CDRs), our method does not rely on it in the generation process. It makes the model computationally efficient while still achieving high sequence recovery and diversity rates as compared to the current baselines. Reprogrammed language BERT model (*ReprogBert*) is our proposed infilling model, where the English language BERT remains unchanged and frozen (source domain), and we introduce additional amino acid embeddings (target domain) together with the linear matrices ($\theta \in \mathbb{R}^{|V_t| \times |V_s|}$ and $\gamma \in \mathbb{R}^{|V_s| \times |V_t|}$) to project from one domain to another. During CDR infilling training, only the projection matrices and protein embeddings are fine-tuned, the language model remains unmodified. The bottom diagram shows the schematic view of the reprogramming: $f_\theta : x_t \to x_s$ is transforming input protein sequence (target domain (T)) into input word sequence (source domain (S)) and $g_\gamma : y_s \to y_t$ reverses the mapping. Thus, for a masked protein sequence $x_t$ we get predicted CDR-infilled antibody $y_t = f_\gamma(M(f_\theta(x_t)))$, where $M$ is the pretrained language model.

There is a high demand and need for efficient in-silico methods for designing CDRs with improved specificity and other desired properties, to reduce the cost and time associate with wet lab production and testing of antibody candidates. Generative machine learning has emerged as an attractive and viable path for this purpose. For example, for a more general task of protein design, creating new protein sequences that fold to a desired 3D structure and/or exhibit a specific function, many deep generative models have been adapted and expanded (Ingraham et al., 2019; Cao et al., 2021; Karimi et al., 2020; Syrlybaeva & Strauch, 2022; Lee & Kim, 2022; Anand & Achim, 2022). However, compared to other protein design challenges, CDR design (Akbar et al., 2022b; Eguchi et al., 2020; Shin et al., 2021; Adolf-Bryfogle et al., 2018; Fu & Sun, 2022; Kong et al., 2022; Luo et al., 2022), especially CDR-H3 design, comes with additional complexities, such as out-of-distribution generation to accommodate functional novelty. Additionally, in antibody design, sequence similarity may not reflect binding behavior. For example, in HER2 binding antibodies, two very similar sequences (Levenshtein distance < 2) had opposing binding behavior (Mason et al., 2021). Furthermore, it is often desirable to explore new antigen binding modes, when designing antibodies for a target of interest. Such out-of-distribution sample generation remains challenging, particularly in a template-constrained generation scenario.

Most of the prior works compromise on the sequence and structural diversity in generated CDRs for high amino acid recovery and low root mean square deviation (RMSD) from ground truth structure. Moreover, the sequence-based models typically involve LLM training from scratch on NGS repertoire (Olsen et al., 2022), or GNN training on a small sample of antibody sequence-structure pairs (Jin et al., 2021). The GNN-based models also come with a cost associated with inference, e.g., iterative design of nodes and edges in a graph via autoregressive decoding.

To address these challenges, we propose an alternative sequence-only framework (see Fig. 1 for an overview), that is reprogramming existing out-of-domain English language BERT model (Devlin et al., 2018) toward the protein infilling task, given the rest of the sequence as a template. We term this model *ReprogBert*. Additionally, for our sequence-based infilling task we also consider in-domain specialized protein model *ProtBert* (Elnaggar et al., 2020) as well as the English language BERT (*EnglishBert*), whose out-of-domain language token embeddings are replaced with in-domain amino acid embeddings (see Fig. 5 in Appendix for details). We compare all of our proposed infilling methods with physics-based and graph-based generative models on a list of tasks ranging from template constrained CDR design to CDR sequences with predicted SARS-COV-2 neutralization ability. We show that while ReprogBert enjoys comparable high structural consistency, and lower sequence perplexity, when matched against the baselines, it shows high amino acid CDR recovery while providing additional benefit regarding generating highly diverse CDR sequences. These results suggest the potential of ReprogBert toward on-demand generation of the out-of-distribution sequences in the learning from limited data scenario. The other proposed baseline systems, EnglishBert and ProtBert, achieve high CDR sequence recovery rates with consistent structural integrity, although having a modest sequence diversity performance.

In summary, in this work we: **(i)** propose ReprogBert, a system for protein sequence infilling using model reprogramming for the task of antibody CDR design, **(ii)** show promising performance results as compared to many baselines (including our own proposed ProtBert and EnglishBert baseline infilling methods) and over multiple benchmarks, where our ReprogBert model upholds structural integrity and sequence recovery, while achieving valuable high diversity of the generated sequences. Moreover, the generated CDR sequences frequently have the lowest perplexity, reflecting their well-formed composition and naturalness. ReprogBert further shows its promise in harder CDR design tasks, can handle multiple CDR infilling at once, and does not need structure template information, and **(iii)** observe high data-efficiency of the reprogrammed model, having only a few training parameters, it can be efficiently trained in the data-scarce domains, such as antibody design, while still leveraging information from large out-of-domain language pretraining.

## 2    REPROGRAMMING FOR PROTEIN SEQUENCE INFILLING

The field of model reprogramming (MR) has focused on repurposing pretrained machine learning (ML) models for varied ML tasks in different domains. It was firstly proposed in an adversarial setting of stealthy resource alternation in (Elsayed et al., 2018) and later extended to cross-domain resource-efficient transfer learning (Chen, 2022; Neekhara et al., 2022). MR achieves state-of-the-art performance in many tasks, especially in data-limited classification settings, including reprogramming general images for bio-medical measurements (Tsai et al., 2020), human voice for time-series (Yang et al., 2021), and sentence sentiment for protein property (Vinod et al., 2020), to name a few. While current MR techniques focus on classification tasks, in our work we seek to extend MR capabilities into generative tasks through reprogramming large pretrained language models for protein sequence infilling. To the best of our knowledge, this work is the first study for such an endeavor.

Given a protein sequence, we propose novel CDR loop design as a form of a template-infilling. The template is provided by the amino acid sequence of the constant region of the antibody, as those are conserved and less likely to change, while the sequences corresponding to CDR can vary and change the structure of the antigen binding interface, resulting into modified antigen affinity and specificity. It should be mentioned though the infilling here is performed to design CDRs of antibodies, the framework can be leveraged to infill any protein sequences.

Figure 1 presents an overview of our proposed framework, *ReprogBert*, the reprogrammed language model for protein sequence infilling. Specifically, we use the pretrained English BERT model (Devlin et al., 2018) (in our experiments, it is the `base-bert-uncased` from HuggingFace) and reprogram it for infilling the CDR part of the antibodies.

The number of tokens in the original language task (i.e., source domain) is denoted by $V_s$ (in our experiments $|V_s| = 30522$ word tokens). The language sentence can then be represented as $y_s = \langle w_1, w_2, \ldots, w_n \rangle$, where $w_i$ is the word token. The number of tokens in the task of interest (i.e., target domain) is denoted by $V_t$ (in our experiments $|V_t| = 30$ protein tokens: 20 amino acid tokens and 10 auxiliary tokens). The protein sentence can then be represented as $x_t = \langle a_1, a_2, \ldots, a_n \rangle$, where $a_i$ is an amino acid token.

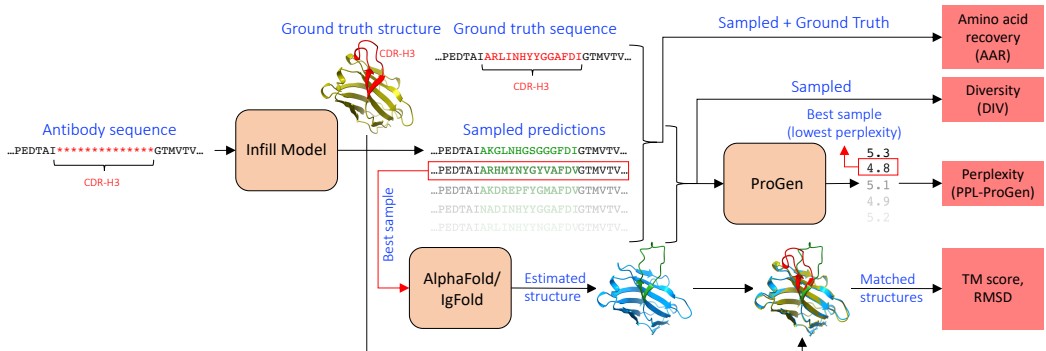

Figure 2: Evaluation process and the computed metrics. For each masked antibody input sequence we generate 100 predicted samples. Amino acid recovery (AAR) is computed for the specific sequence region (e.g., CDR-H3), measuring the fraction of exact matches between ground truth and the sampled sequences. Diversity (DIV) uses only the generated samples to compute the complement of the average recovery for all pairwise comparisons in the set (the higher the number, the more dissimilar is each sample to all the others). Perplexity (PPL-ProGen) is computed as the average of all the sampled sequences (masking only the region of interest), using off-the-shelf autoregressive Transformer protein model ProGen (Nijkamp et al., 2022), which reflects "naturalness" of the designed sequences. The sample with the minimum perplexity (red box with an arrow) is then used for 3D structure prediction using AlphaFold (Jumper et al., 2021) or IgFold (Ruffolo et al., 2022) models and compared with ground truth to compute template modeling (TM) score (Zhang & Skolnick, 2004) and the root mean squared deviation (RMSD) from the input structure.

We define two mappings (see bottom plot in Fig. 1) $f_\theta : x_t \rightarrow x_s$, transforming input protein sequence into input word sequence and $g_\gamma : y_s \rightarrow y_t$, reversing the transformation by mapping output word sequence into protein one. Following the approach in (Elsayed et al., 2018; Tsai et al., 2020; Vinod et al., 2020) we constrain these mappings to be linear transformations between the source and target domains. In other words, these mappings are represented as: $x_s = x_t\theta$ and $y_t = y_s\gamma$, where the linear projection matrices $\theta \in \mathbb{R}^{|V_t| \times |V_s|}$ and $\gamma \in \mathbb{R}^{|V_s| \times |V_t|}$ are the parameters of the transformations. During training, all model parameters are fixed and only $\theta$ and $\gamma$ are optimized. Specifically, we update $\theta$ and $\gamma$ with respect to minimizing $\mathcal{L}_{NLL}(y_t, y_t^*)$, the loss between the estimated infilled protein sequence $y_t = f_\gamma(M(f_\theta(x_t)))$, given the CDR-masked anitbody $x_t$ and the ground truth sequence $y_t^*$.

## 3 EXPERIMENTS

In this section we present evaluation results of our proposed methods on template constrained CDR design using Structural Antibody Database (SabDab) (Dunbar et al., 2013) and Rosetta Antibody Design (RabD) (Jin et al., 2021), as well as SARS-CoV2 (CoV-AbDab dataset (Raybould et al., 2021)) neutralization using the model's generated antibodies. In what follows, we first discuss the evaluation metrics, followed by the introduction of the baseline models and the presentation of the results on three datasets.

### 3.1 EVALUATION METRICS

For each input protein sequence in our experiments we generated 100 samples using our infill models. To measure the quality of these samples, we then compute the following evaluation metrics (see Fig. 2 for an illustration). *Amino acid recovery (AAR)* is computed for the specific sequence region of interest (e.g., CDR-H3), measuring the percent of the exact matches between ground truth and the sampled sequences. The range is 0-100, and the higher the AAR, the more accurate the recovery. *Diversity (DIV)*, on the other hand, uses only the sampled proteins to compute the complement of the average recovery of all pairwise comparisons in the set. Here the range is 0-100 and the higher the number, the more dissimilar are samples among themselves. While in general it holds true that the recovery and diversity are inversely correlated, i.e., higher recovery rate leads to lower diversity,

and vice versa, CDR design calls for generative models that achieve at least above 30% recovery (Weitzner et al., 2015), while at the same time are able to maintain high sequence diversity.

For *perplexity* (the model's predicted probabilities for every residue in a given sequence) we use off-the-shelf autoregressive Transformer protein model ProGen (Nijkamp et al., 2022) to compute *PPL-ProGen* as the average of 100 samples (masking only the region of interest). Specifically, we used ProGen2-small (151M parameters), which has been pretrained on the mixture of Uniref90 (Suzek et al., 2015) and BFD30 (Steinegger & Söding, 2018) datasets. For perplexity, the lower values mean the better performance, indicating stronger "naturalness" of generated CDRs. The sampled protein sequence with the minimum perplexity is then used for 3D structure prediction using protein folding model (e.g., AlphaFold (Jumper et al., 2021) or IgFold (Ruffolo et al., 2022)). The full predicted and ground truth structures are then compared to compute *template modeling (TM)* score (Zhang & Skolnick, 2004) (range 0-100, higher the better) and the *root mean squared deviation (RMSD)* (lower the better), focusing only on the CDR part. The suffix AF in the metric names represents AlphaFold, while IF means IgFold.

## 3.2 BASELINE MODELS

We included the following baseline methods to compare against our BERT-based infilling models. *LSTM* from (Saka et al., 2021) and (Akbar et al., 2022a), which, similar to ours, is a sequence-only model, however of smaller capacity, having a single attention layer between the input and output layers. *AR-GNN* - autoregressive graph neural network (Jin et al., 2021), which is a sequence and structure-based model, at each step first it predicts the amino acid, followed by the edge generation between the current and all the past residues. *RefineGNN* (Jin et al., 2021) is a model that designs protein sequence and 3D structure of CDR together as graphs. At each step the method predicts residues autoregressively and simultaneously refines the predicted global structure, which in turn helps in subsequent residue prediction. To improve computational efficiency, they employ coarse-grained modeling by clustering every predefined number of context residues in a block, thus reducing the size of the computational graph.

| CDR | Train | Validation | Test | Average CDR length | Average CDR diversity |
|-----|-------|------------|------|--------------------|-----------------------|
| CDR-H1 | 4050 | 359 | 326 | 8.1 | 60.8 |
| CDR-H2 | 3876 | 483 | 376 | 7.9 | 68.2 |
| CDR-H3 | 3896 | 403 | 437 | 14.5 | 76.9 |

Table 1: Statistics of the Structural Antibody Database (SabDab) for the training, validation and test splits across the three CDRs. We also show the average number of amino acids per CDR and average CDR diversity (length-normalized) across proteins. As can be seen CDR-H3 is the longest and most diverse and therefore represents the most challenging prediction task.

| | | SabDab CDR-H1 | | | | | | | | |
|-----|-----|-----|-----|-----|-----|-----|-----|-----|-----|-----|
| | PPL | PPL-ProGen | RMSD | RMSD-AF | RMSD-IF | TM-AF | TM-IF | AAR | AAR>30% | DIV |
| LSTM | 6.79 | – | – | – | – | – | – | – | – | – |
| AR-GNN | 6.47 | – | 2.97 | – | – | – | – | – | – | – |
| Refine-GNN | 6.09 | 3.5 | 1.18 | 4.42 | 1.78 | 84.0 | 93.6 | 61.2 | yes | 47.3 |
| ProtBert | – | 3.5 | – | 4.16 | 1.68 | 84.4 | 93.8 | 64.7 | yes | 4.6 |
| EnglishBert | – | 3.7 | – | 4.22 | 1.67 | 84.1 | 93.8 | 63.6 | yes | 5.8 |
| ReprogBert | – | 3.3 | – | 4.31 | 1.73 | 84.0 | 93.7 | 56.0 | yes | 29.1 |

Table 2: Evaluation results on the SabDab dataset for CDR-H1 in the heavy chain. Dark grey cell denote best results, while light grey are the second best. ReprogBert generates sequences with lowest perplexity, second best diversity and high enough AAR and structural consistency. RefineGNN yields the best diversity. ProtBert and EnglishBert, both lack in the CDR sequence. RMSD-AF and RMSD-IF, the structural consistency metric based on the AlphaFold and IgFold predicted structures, respectively, shows similar performance across all the methods (same for TM scores). However, AlphaFold-based values show more structure inconsistencies as compared to IgFold. This is likely due to IgFold being specifically trained on the antibody domain, while AlphaFold is a more general framework, thus introducing certain errors in the structure estimation.

| | | | | | SabDab CDR-H2 | | | | | |
|---|---|---|---|---|---|---|---|---|---|---|
| | PPL | PPL-ProGen | RMSD | RMSD-AF | RMSD-IF | TM-AF | TM-IF | AAR | AAR>30% | DIV |
| LSTM | 7.21 | – | – | – | – | – | – | – | – | – |
| AR-GNN | 6.86 | – | 2.27 | – | – | – | – | – | – | – |
| Refine-GNN | 6.58 | 3.4 | 0.87 | 3.05 | 1.40 | 85.7 | 93.9 | 48.9 | yes | 38.7 |
| ProtBert | – | 3.6 | – | 3.10 | 1.32 | 85.6 | 93.9 | 59.5 | yes | 5.5 |
| EnglishBert | – | 4.0 | – | 3.07 | 1.32 | 85.6 | 93.9 | 59.1 | yes | 7.7 |
| ReprogBert | – | 3.9 | – | 3.02 | 1.40 | 85.8 | 93.8 | 53.0 | yes | 37.9 |

Table 3: Evaluation results on the SabDab dataset for CDR-H2 in the heavy chain. As compared to Table 2, all of our proposed infill methods now outperform RefineGNN in terms of AAR metric, while reprogBert also provides second best diversity.

| | | | | | SabDab CDR-H3 | | | | | |
|---|---|---|---|---|---|---|---|---|---|---|
| | PPL | PPL-ProGen | RMSD | RMSD-AF | RMSD-IF | TM-AF | TM-IF | AAR | AAR>30% | DIV |
| LSTM | 9.20 | – | – | – | – | – | – | – | – | – |
| AR-GNN | 9.44 | – | 3.63 | – | – | – | – | – | – | – |
| Refine-GNN | 8.38 | 7.2 | 2.50 | 5.62 | 3.43 | 85.0 | 94.0 | 28.2 | no | 25.7 |
| ProtBert | – | 6.8 | – | 5.40 | 3.39 | 85.2 | 94.0 | 41.5 | yes | 14.5 |
| EnglishBert | – | 5.9 | – | 5.53 | 3.26 | 84.9 | 94.0 | 35.6 | yes | 59.8 |
| ReprogBert | – | 5.4 | – | 5.54 | 3.44 | 85.1 | 94.0 | 32.6 | yes | 67.4 |

Table 4: Evaluation results on the SabDab dataset for CDR-H3. As compared to CDR-H1 ( Fig. 2) and CDR-H2 (Fig. 3), longer CDR-H3 design is more challenging, which shows a drop in AAR across all the methods. ReprogBert clearly outperforms RefineGNN on this hard task, as evident from lower PPL, better AAR, and better diversity.

## 3.3 EXPERIMENTS ON THE STRUCTURAL ANTIBODY DATABASE (SABDAB)

SabDab (Dunbar et al., 2013) is a dataset containing antibody sequences and the corresponding 3D structure information, annotated with several properties like gene details, heavy and light chain pairings, CDR location, etc. For this experiment, we used the dataset curated by (Jin et al., 2021) and the statistics are shown in Table 1. The evaluation results are shown in Tables 2, 3, and 4. We note that the values for PPL and RMSD metrics for LSTM, AR-GNN and Refine-GNN are from the published results (Jin et al., 2021). Comparing across the three experiments, we can see that CDR-H1, CDR-H2 and CDR-H3 estimations are progressively harder problems, which is reflected in the drop of AAR across all the methods. Among the proposed infill methods, ProtBert achieves the highest AAR across all experiments. We also can see that ReprogBert has a good recovery accuracy and at the same time generates very diverse CDR sequences. We emphasize that this performance is without the access to the available 3D structure information. RefineGNN, on the other hand, using both sequence and structure constraints, overall preforms competitively, generating CDR sequences that are accurate and diverse. Nevertheless, the advantage of ReprogBert is more prominent for longer CDR-H3, which is the hardest design task of all three, where ReprogBert evidently outperforms RefineGNN in term of perplexity, AAR, and diversity, while maintaining structural integrity. Finally, in Fig. 5 we show the results of all three CDRs infilling at once. Our BERT-based models are not architecturally limited to a single CDR generation, therefore can infill multiple regions at once with similar high recovery, structural consistency, and diversity scores.

Since our BERT-based infill models do not estimate protein structure, we use AlphaFold (Jumper et al., 2021) and IgFold (Ruffolo et al., 2022) to estimate 3D structure from the generated sequence and compute TM and RMSD scores with respect to groundtruth native structure. We can see from the Tables 2, 3, 4, and 5 that all the methods have similar structural consistency results (TM and RMSD-AF). However, these values are consistently higher when compared to RMSD for "natively" predicted structure (AR-GNN and Refine-GNN), which is likely due to the estimation errors introduced by the AlphaFold or IgFold algorithm. Since RefineGNN focuses on recovering both groundtruth sequence and structure, it does so by sacrificing exploration of the broader sequence space accessible to a given structure (Tian & Best, 2017), which is not the case for ReprogBert.

To further qualitatively illustrate the effect of recovery and diversity on the sampled sequences, we show in Fig. 3 AlphaFold-generated 3D structures of the protein sequences generated by the

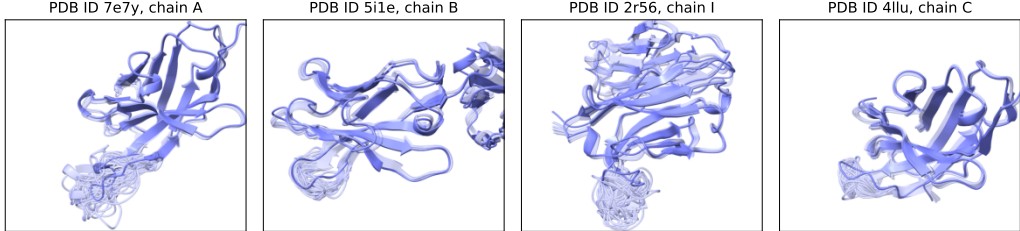

Figure 3: AlphaFold-estimated 3D structures of the proteins generated by the ReprogBert model on SabDab dataset. Each plot shows 30 generated samples for a specific PDB ID, where the CDR-H3 part of the input has been masked and the model then generates CDR-H3 sequence. The ground truth and the generated CDR are shown on the bottom part of each figure using solid and faded colors, respectively. As can be seen, CDR-H3 part shows high structural diversity, confirming the same findings as in Table 4, i.e., that ReprogBert achieves high recovery rate while maintaining the highest sequence diversity.

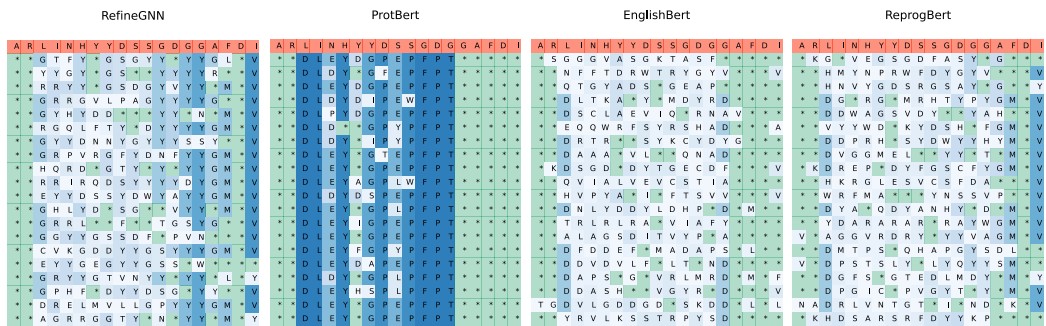

Figure 4: Visualization of the sequence recovery and diversity metrics for generated CDR-H3 (PDB ID 7e7y) across different models. The top row colored in red shows the ground truth CDR-H3 sequence, while the following 20 rows show the same for the generated CDR-H3s. The green cell with the star symbol represents the same amino acid as in the ground truth, while the white/blue cell shows new and different generated residues. The darker shade of the blue cell represents the frequency of the amino acid in that column.

ReprogBert model. High structural diversity of the CDR-H3 is clearly visible by the coverage of the CDR-H3 ensemble (ground-truth shown using opaque while generated shown as transparent) . Fig. 4 presents a visualization of sequence similarity (in green)/diversity (in white to blue) across models. For example, for ProtBert the third column has a residue D in all the rows (high frequency), thus having the darkest shade, while for ReprogBert the last column has only two generated Y's (low frequency), thus colored in the light shade of blue. Therefore, the method with the high recovery and high diversity rates will have many green and light blue cells. Comparing with Table 4, we indeed see that ReprogBert has highest diversity represented by the largest number of light blue cells, at the same time ProtBert has most green cells (highest AAR), but also many dark blue cells (low diversity). It can also be seen that RefineGNN has lower diversity and lower recovery, as compared to ReprogBert. Further, the 2D kernel density plot as a function of isoelectric point (pH when net charge is 0) and length of CDR-H3 shown in Figure 8 in Appendix implies ReprogBert maintains highest physicochemical similarity to the natural CDRs.

### 3.4 ANTIGEN-SPECIFIC ANTIBODY DESIGN

The goal here is to design a CDR such that it binds a given antigen, given the antibody sequence template. For this experiment, we used the dataset curated by (Jin et al., 2021), statistics of which is shown in Table 6. In particular it consists of all the SabDab 6 for training, excluding sequences in the same cluster as test antibodies, which were proposed by (Adolf-Bryfogle et al., 2018). In addition to

| | SabDab CDR-H1,2,3 | | | | | | | |
|---|---|---|---|---|---|---|---|---|
| | PPL-ProGen | RMSD-AF | RMSD-IF | TM-AF | TM-IF | AAR | AAR > 30 | DIV |
| ProtBert | 4.9 | 4.8 | 2.62 | 85.0 | 94.3 | 57.3 | yes | 8.2 |
| EnglishBert | 5.2 | 4.83 | 2.62 | 85.0 | 94.2 | 56.3 | yes | 8.3 |
| ReprogBert | 3.9 | 4.95 | 2.73 | 84.8 | 94.0 | 42.4 | yes | 57.4 |

Table 5: Evaluation results on the three heavy chain CDR loops generated at once using the SabDab dataset. This is the most challenging task compared to designing one CDR at a time. However, since our BERT-based models are not architecturally limited to a single CDR generation, they can infill multiple protein regions at once with similar high recovery scores, as opposed to AR-GNN and RefineGNN. Moreover, the reprogrammed model showed the lowest perplexity, good structural consistency, and the highest sequence variability among the three proposed methods.

| CDR | Train | Validation | Test | Average CDR length |
|---|---|---|---|---|
| CDR-H3 | 8646 | 98 | 58 | 14.5 |

Table 6: Statistics of the Rosetta Antibody Design (RabD) dataset for CDR-H3.

the earlier defined baselines, for this experiment, similar to (Jin et al., 2021), we compared against a physics-based baseline, RabD (Adolf-Bryfogle et al., 2018), which first grafts a CDR from an internal database into the groundtruth antibody structure, followed by iterations of amino acid substitutions and energy minimization. The results are shown in Table 7. The values for PPL, RMSD, and AAR metrics for RabD, LSTM, AR-GNN and Refine-GNN baselines are from the published results in (Jin et al., 2021). Our proposed BERT-based infill methods outperform all the baselines in both the accuracy of the reconstruction as well as the diversity of generation. ProtBert achieves the highest AAR score, while the ReprogBert has the best diversity rate with AAR comparable to RefineGNN.

| | RabD CDR-H3 | | | | | | | | | |
|---|---|---|---|---|---|---|---|---|---|---|
| | PPL | PPL-ProGen | RMSD | RMSD-AF | RMSD-IF | TM-AF | TM-IF | AAR | AAR > 30 | DIV |
| RabD | 9.20 | – | – | – | – | – | – | 28.53 | no | – |
| LSTM | 9.20 | – | – | – | – | – | – | 22.53 | no | – |
| AR-GNN | 9.44 | – | 3.63 | – | – | – | – | 23.86 | no | – |
| Refine-GNN | 8.38 | 4.7 | 2.50 | 5.06 | 2.52 | 82.9 | 96.0 | 35.4 | yes | 31.1 |
| ProtBert | – | 7.7 | – | 5.42 | 2.35 | 82.3 | 96.2 | 53.1 | yes | 11.6 |
| EnglishBert | – | 7.8 | – | 5.34 | 2.19 | 82.4 | 96.3 | 54.9 | yes | 10.1 |
| ReprogBert | – | 5.1 | – | 4.72 | 2.47 | 83.0 | 96.1 | 36.3 | yes | 62.1 |

Table 7: Evaluation results on the RabD dataset for CDR-H3. Our infilling models outperform RefineGNN, with ProtBert achieving the highest AAR score, while the ReprogBert has the best diversity rate with AAR comparable to RefineGNN. As before, ProtBert and EnglishBert show better recovery performance but suffer from less diverse generation.

## 3.5 CORONAVIRUS ANTIBODY DATABASE (COV-ABDAB)

| CDR | Train | Validation | Test | Average CDR length |
|---|---|---|---|---|
| CDR-H3 | 2282 | 291 | 291 | 15.7 |

Table 8: Statistic of Coronavirus Antibody Database (CoV-AbDab) dataset for CDR-H3.

CoV-AbDab (Raybould et al., 2021) is a public database documenting all published and patented antibodies and nanobodies able to bind to coronaviruses, including SARS-CoV2 and SARS-CoV1. We used the dataset curated by (Jin et al., 2021) from CoV-AbDab, whose statistics is shown in Table 8. The evaluation results are shown in Table 9, where only the sequence-based metrics are presented since the ground truth structure information is unavailable for this task. The results are presented for the case of training only on CoV-AbDab and the case of training on both CoV-AbDab and SabDab datasets, showing overall similar trend, with ProtBert being the most accurate in AAR evaluation,

while ReprotBert achieving the highest diversity while maintaining good sequence recovery and low perplexity.

| | Training on CoV-AbDab | | | | Training on CoV-AbDab + SabDab | | | |
|---|---|---|---|---|---|---|---|---|
| | PPL-ProGen | AAR | AAR > 30 | DIV | PPL-ProGen | AAR | AAR > 30 | DIV |
| ProtBert | 6.0 | 50.7 | yes | 13.6 | 7.8 | 49.6 | yes | 10.7 |
| EnglishBert | 6.3 | 49.3 | yes | 9.5 | 8.2 | 49.2 | yes | 11.0 |
| ReprogBert | 5.7 | 39.3 | yes | 60.2 | 4.9 | 37.7 | yes | 64.1 |

Table 9: Evaluation results on the CoV-AbDab dataset for generated CDR-H3. Since no ground truth structure is available for this dataset, the other structure consistency metrics are not computed.

The second step of our evaluation is to measure the ability of the generated antibodies to neutralize SARS-CoV2 virus, for which we follow the setup of (Jin et al., 2021). Specifically, we employ the neutralization classifier, composed of SRU encoder (Lei, 2021), pooling and feed-forward network, as provided in (Jin, 2022), together with the iterative target augmentation (ITA) framework (Yang et al., 2020). The goal is to additionally fine-tune the infilling models to generate CDRs resulting into better neutralizing antibodies, as measured by the classifier. Table 10 presents the results. Note that the performance values for the neutralization classifier, LSTM, AR-GNN and Refine-GNN are from the published results in (Jin et al., 2021), for which they pretrained these models on SabDab dataset followed by the training on CoV-AbDab. As can be seen from the table, under both training scenarios, our ReprogBert infilling method gets the largest improvement over the original neutralization classifier, achieving 75.6 % and 76.7 % neutralization scores, respectively.

| | Neutralization Score | |
|---|---|---|
| Model | CoV-AbDab | CoV-AbDab + SabDab |
| Original | – | 69.3 |
| LSTM | – | 72.0 |
| AR-GNN | – | 70.4 |
| Refine-GNN | – | 75.2 |
| ProtBert | 72.7 | 74.7 |
| EnglishBert | 70.5 | 71.0 |
| ReprogBert | 75.6 | 76.7 |

Table 10: Neutralization of SARS-CoV-2 virus as predicted by the pre-trained SARS-CoV-1 / SARS-CoV-2 classifier. The neutralization score is defined as the predicted probability of a given antibody to neutralize the SARS-CoV-2 virus, as measured by the neutralization classifier.

## 4    CONCLUSION

In this work we introduced Reprogramming for Protein Sequence Infilling, ReprogBert, a framework leveraging pretrained language models for protein sequence infilling. Specifically, we formulated variable CDR loop design as a template-infilling, where the template is provided by the constant region of the antibody. Results show promising performance, when compared to existing sequence and graph-based deep generative baselines over multiple benchmarks, where our ReprogBert model upholds structural integrity, sequence recovery, and naturalness, while achieving high novelty and diversity of the generated sequences. The improvement is more obvious for the longer CDR-H3. ReProgBert can handle multiple CDR infilling at once with losing performance. Generated antibodies also show antigen specificity and improved virus neutralization. Finally, it is worth emphasizing the high data-efficiency of the reprogrammed model, which results from having only a few training parameters (consisting of two linear projection matrices) that can be efficiently trained in the data-scarce domains, such as antibody design, while still leveraging information from large out-of-domain language pretraining. This advantage allows the sequence-based reprogrammed model to perform competitively or better with respect to other BERT-based models or baselines that learn from both sequences and structures.

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

## A    RELATED WORK ON PROTEIN DESIGN

Protein design involves the design of new protein sequences that fold to a desired 3D structure and/or exhibit a specific function. Computational techniques for designing novel and diverse proteins are an active area of research. Physics based methods that rely on energy minimization have been proposed for designing general proteins (Leaver-Fay et al., 2011; Huang et al., 2011), as well as specifically for antibodies (Pantazes & Maranas, 2010; Li et al., 2014; Adolf-Bryfogle et al., 2018), but these are computationally expensive. Recently, generative deep learning techniques like Generative Adversarial Networks (Goodfellow et al., 2020), Variational Autoencoders (Kingma & Welling, 2013), Graph Neural Networks (Scarselli et al., 2008; Gilmer et al., 2017), autoregressive language models (LSTM and Transformer based) (Vaswani et al., 2017), and diffusion based models (Ho et al., 2020) have been used for protein and antibody design (Wang et al., 2018; Akbar et al., 2022b; Amimeur et al., 2020; Eguchi et al., 2020; Shin et al., 2021; Kong et al., 2022; Fu & Sun, 2022; Syrlybaeva & Strauch, 2022; Lee & Kim, 2022; Anand & Achim, 2022). Some representative works are discussed below. (Ingraham et al., 2019) and (Cao et al., 2021) proposed a graph and a multimodal transformer based model, respectively, for designing proteins conditioned on the backbone structure/fold. (Karimi et al., 2020), developed a guided conditional Wasserstein Generative Adversarial Networks (gcWGAN) for fold based protein design. Another method that uses GANs to generate a distance matrix representation of proteins from which 3D coordinates can be recovered was proposed by (Anand & Huang, 2018). Variational autoencoder based methods have also been proposed for conditional generation of protein sequences (Greener et al., 2018; Das et al., 2021) and for direct generation of 3D coordinates of immunoglobulin proteins (Eguchi et al., 2020).

Several of the above-mentioned architectures have been extended to the specific problem of antibody design, which is considered challenging due to focus on designing long, variable, and unstructured CDRs. (Melnyk et al., 2021) provides benchmarking of several deep generative models on antibody design. Recently, (Jin et al., 2021) proposed an iterative refinement graph neural network for jointly designing the sequence and 3D structure of the CDR regions of antibodies for improving its properties. A deep generative model that jointly models sequences and structures of CDRs based on diffusion processes and equivariant neural networks has been proposed in (Luo et al., 2022). A geometry-constrained energy-based model has been suggested by (Fu & Sun, 2022).

Other approaches for protein design include modeling it as a constraint satisfaction problem (Strokach et al., 2020), equivariant 3D translation (Kong et al., 2022) and by using combinatorial bayesian optimization (Khan et al., 2022).

## B    OVERVIEW OF PROPOSED BASELINE MODELS

Figure 5 shows diagrams of the proposed baseline BERT-based infilling models: ProtBert, a specialized model that has been pretrained on millions of protein sequences and EnglishBert, the traditional English language model, where we replaced word embeddings with new learnable amino acid embeddings. Similar as our main proposed method, ReprogBert, these two models are sequence-only methods and they use maskings to infill the regions of interest.

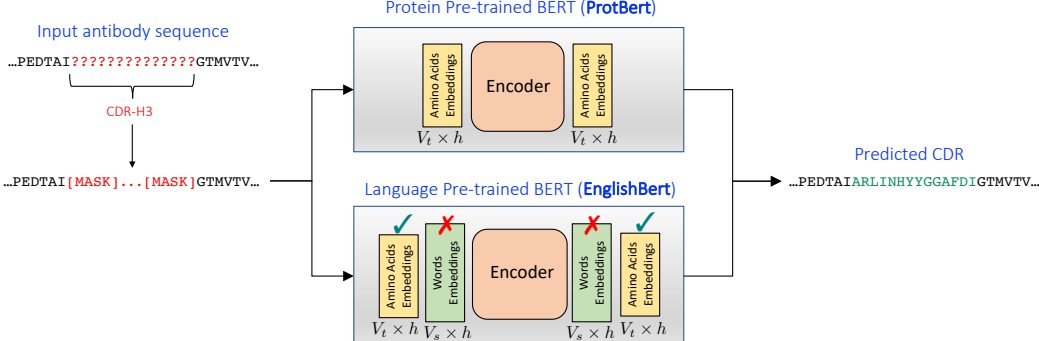

Figure 5: Baseline methods proposed for protein sequence infilling. Given an input antibody sequence, where part of the amino acids is missing (e.g., CDR-H3), the goal is to infill them using information from the rest of the protein. The infilling problem is formulated similar to the masked-language modeling task, where the missing amino acids are marked with a token ⟨MASK⟩ and the model generates amino acids token to infill them. These are sequence-only methods and do not rely on any structure information during generation process. The top diagram shows *ProtBert*, the BERT model that has been pretrained on the protein sequences and therefore can be applied to the protein infilling task as is (the entire model is still fine-tuned on the downstream infilling task). The bottom diagram shows traditional English language BERT model (*EnglishBert*), whose incompatible word embeddings ($V_s \times h$, $V_s$ is the number of language tokens, $h$ - latent model dimension) are swapped with the trainable amino acid embeddings ($V_t \times h$, $V_t$ is the number of amino acid tokens). The full model is then fine-tuned on the infilling dataset.

## C   MODEL ARCHITECTURE AND TRAINING

In Table 11 we present the architectural details of our BERT-based models for the protein sequence infilling, while Table 12 shows the settings used for model training.

| Model | Number of parameters | Number of layers | Hidden layer size | Number of heads | Vocab size | Pretraining Data | Reference |
|---|---|---|---|---|---|---|---|
| ProtBert | 420M | 30 | 1024 | 16 | 30 | BFD100 (572 GB, 2 bil proteins) Uniref100 (150 GB, 216 mil proteins) | (Devlin et al., 2018) |
| EnglishBert / ReprogBert (based on HF bert-base-uncased) | 110M | 12 | 768 | 12 | 30522 (english) 30 (protein) | English Wikipedia (40 GB, 6.5 mil sentences) BookCorpus (6 GB, 74 mil sentences) | (Elnaggar et al., 2020) |

Table 11: Architectural details of the BERT-based model for protein sequence infilling. Note that for ReprogBert the number of trainable parameters is defined by the two $\mathbb{R}^{30522 \times 30}$ matrices.

| Learning rate | Batch size | Optimizer |
|---|---|---|
| $1e^{-5}$ | 32 | Adam |

Table 12: Training details for ProtBert, EnglishBert and ReprogBert. For example, for SabDab dataset to reach the best performance it took 5 hours for ReprogBert, 6 hours for EnglishBert and 14 hours for ProtBert, which is equivalent to approximately 1800 epochs (134 minibatch iterations per epoch). Average inference time per protein sequence is 0.02 seconds for ProtBert, and 0.008 seconds for ReprogBert and EnglishBert (as measured on the test set of SabDab for CDR-H3 infilling). For reference, the average inference time for RefineGNN is 0.004 seconds, which is comparable to our ReprogBert.

# D   ABLATION ON DATA

In Table 13 we show an ablation results on the effect of training data size on model performance.

| | SabDab-H3 | | |
| --- | --- | --- | --- |
| Training data fraction | PPL-ProGen | AAR | DIV |
| **ProtBert** | | | |
| 1.0 | 6.8 | 41.5 | 14.5 |
| 0.8 | 6.7 | 41.3 | 13.1 |
| 0.6 | 6.6 | 40.9 | 15.9 |
| 0.4 | 6.4 | 40.5 | 18.9 |
| 0.2 | 6.6 | 40.3 | 18.4 |
| **EnglishBert** | | | |
| 1.0 | 5.9 | 35.9 | 59.8 |
| 0.8 | 5.9 | 35.1 | 57.9 |
| 0.6 | 6.5 | 34.2 | 59.6 |
| 0.4 | 6.4 | 33.6 | 61.4 |
| 0.2 | 6.5 | 33.1 | 63.5 |
| **ReprogBert** | | | |
| 1.0 | 6.0 | 32.6 | 67.4 |
| 0.8 | 5.9 | 32.1 | 67.6 |
| 0.6 | 6.1 | 31.6 | 68.2 |
| 0.4 | 6.3 | 30.8 | 69.5 |
| 0.2 | 6.5 | 29.9 | 70.7 |

Table 13: Ablation results on the effect of training data size on model performance. The fractions 1.0, 0.8, 0.6, 0.4 and 0.2 representing progressively smaller subsets of the original SabDab training dataset. It can be seen that as the size of training data drops, the recovery rate also decreases, while the diversity increases (this is expected as now the generated sequences are less accurate). However, for ProtBert, the decrease is slower, likely due to this model being pretrained on large protein dataset, thus retaining its prediction capacity.

# E   RECOVERY AND DIVERSITY METRICS

Figures 6 and 7 show additional visualizations of the recovery and diversity metrics for CDR-H3 across different methods.

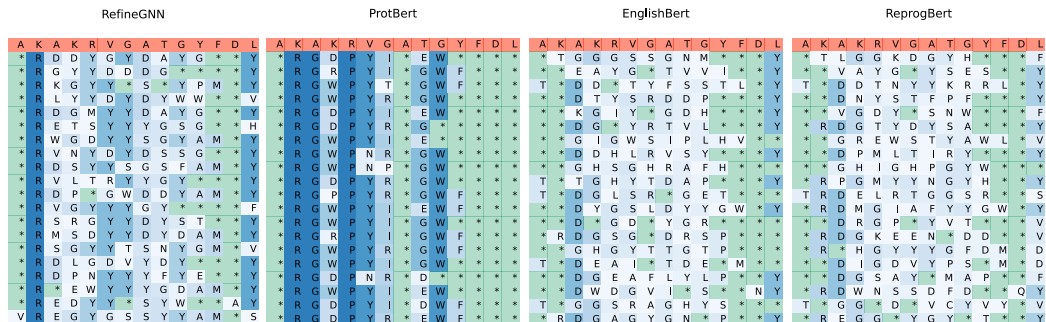

Figure 6: Visualization of the recovery and diversity metrics for CDR-H3 (PDB ID 2r56) across different models. The top red line shows the ground truth CDR-H3 sequence, while the next lines show the generated CDR-H3 by each of the model. The green cell with the star symbol represents the same amino acid as in the ground truth, while the blue cell shows new and different generated residues. The shade of the blue cell represents the frequency of the amino acid in that column. We see that ReprogBert has highest diversity represented by the largest number of light blue cells, at the same time ProtBert has most green cells (highest AAR), but also many dark blue cells (low diversity). It can also be seen that RefineGNN has lower diversity and lower recovery, as compared to ReprogBert.

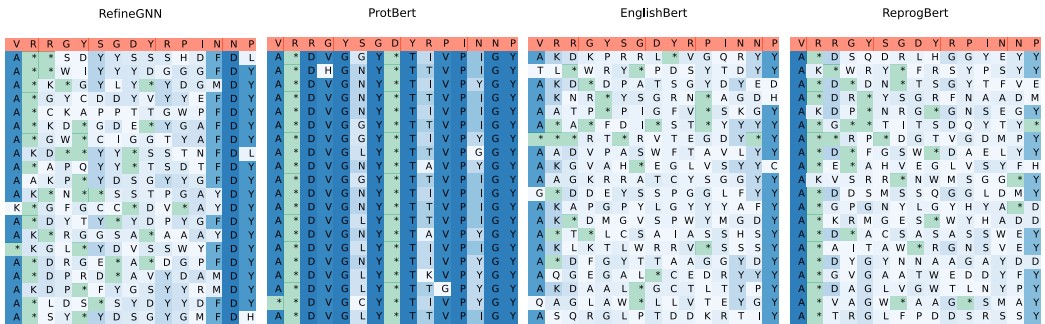

Figure 7: Visualization of the recovery and diversity metrics for CDR-H3 (PDB ID 5y7z) across different models. The top red line shows the ground truth CDR-H3 sequence, while the next lines show the generated CDR-H3 by each of the model. The green cell with the star symbol represents the same amino acid as in the ground truth, while the blue cell shows new and different generated residues. The shade of the blue cell represents the frequency of the amino acid in that column. We see that ReprogBert has highest diversity represented by the largest number of light blue cells, at the same time ProtBert has most green cells (highest AAR), but also many dark blue cells (low diversity). It can also be seen that RefineGNN has lower diversity and lower recovery, as compared to ReprogBert.

## F  PHYSICOCHEMICAL PROPERTY COMPARISON

Finally, in Figure 8 we show 2D kernel density plot as a function of isoelectric point and length of generated CDR-H3 sequences on the test set of SabDab dataset.

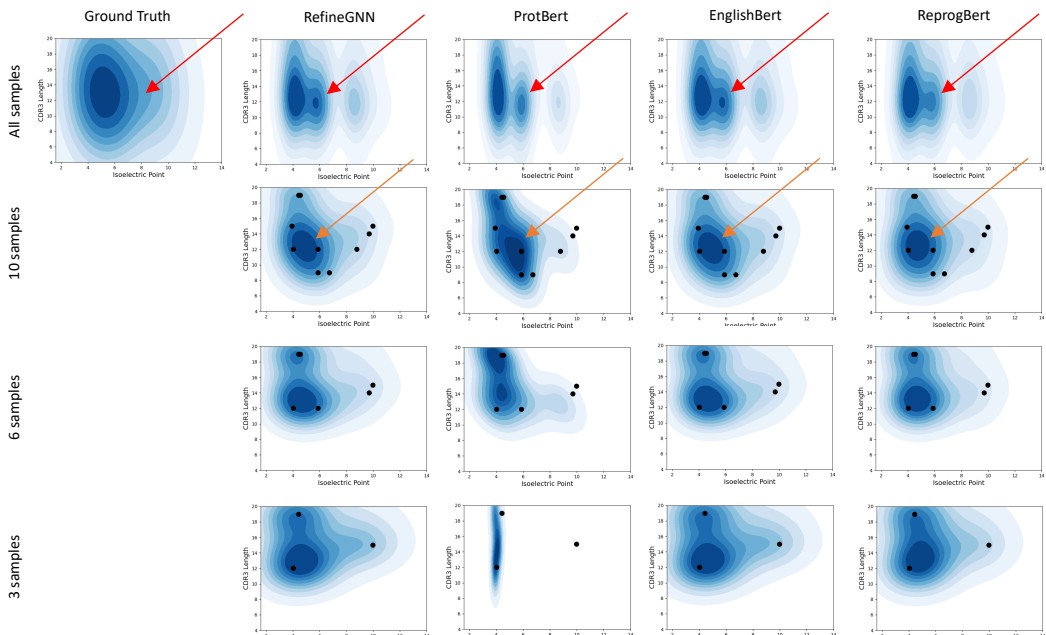

Figure 8: 2D kernel density plot as a function of isoelectric point, the pH at which a particular molecule carries no net electrical charge, and length of generated CDR-H3 sequences on the test set of SabDab dataset. Black dots indicate the ground truth CDR-H3. The top row shows the density of all the sequences, while the following rows show the density for 10, 6 and 3 samples. It can be seen that the ground truth density of all the sequnces (top left corner) has one pronounced peak (for the CDR3 length 13 and pH 5) and another smaller increase of density marked with red arrow. Comparing this region across other models, we see that ReprogBert has the closes resemblance to the ground truth, while others place too much weight there. The second row from the top shows the density for 10 protein sequences, where visual inspection of the region marked with orange arrow reveals that ReprogBert has closest similarity to the ground truth based on the distribution and orientation of the highly dense region, while for other methods theshape of this region is tilted and a second minimum appears.

