# OpenReview forum: "Reprogramming Large Pretrained Language Models for Antibody Sequence Infilling"
_ICLR.cc/2023/Conference — Submitted to ICLR 2023_

### Official Review · Reviewer_qyah · 2022-10-21

**Confidence:** 3
**Correctness:** 2
**Technical Novelty And Significance:** 3
**Empirical Novelty And Significance:** 2
**Recommendation:** 5

**Clarity, Quality, Novelty And Reproducibility:**

It is a novel and original approach that has seen no prior application in biological sequence design. The paper lacks clarity as to why exactly model reprogramming should be the right choice for this biological problem.

**Strength And Weaknesses:**

+The idea is novel
- The performance compared to the baselines is not convincing. The comparably strong performance of EnglishBert, which should be expected to have no knowledge of the problem whatsoever, is especially worrying.
- Not clear how useful sequence diversity is as a quality metric of the model
- It is not clear if the baseline models are fair, as they have fewer learnable parameters.
- Discussion not addressing the implications of the results.

Questions
How do the authors explain the strong performance of EnglishBERT, which has no knowledge about the antibody inpainting problem?
What is the meaning of ProGen perplexity for CDR loops? Do native CDRs have lower perplexity?
Why was EnglishBERT chosen for reprogramming, when ProtBERT is pretrained on a domain much closer to the task?
Why are BERT models pretrained on antibody sequences not considered?

Comments
Figure 1 is confusing. Showing the 3D structure and the light chain sequence on the input side of the figure is distracting, and only upon reading it becomes clear that those are not part of the model input.

**Summary Of The Paper:**

The work proposes to adapt pretrained language models for CDR loop infilling. Essentially, this is a task that requires the model to predict the identity of a span of masked out tokens in a sequence of amino acid tokens. The work proposes to do the adaptation using a model reprogramming approach, where two linear layers are used to map from the (task_input, pretrained_output) domains to the (pretrained_input, task_output) domains, thereby learning a lightweight network that casts the task at hand into something that can be solved by the pretrained model.
This is a novel idea to practitioners in the protein deep learning field, where typically no benefit is expected from using models pretrained on human language. The paper claims that the method achieves good infilling (=reconstruction) performance while also having high sample diversity.


**Summary Of The Review:**

I am not convinced that this method works well. Essentially, the paper claims that the task of antibody sequence infilling can be solved by reprogramming (=adding two linear layers) a BERT model that is pretrained on english language and that this works better than a) a BERT model pretrained on proteins and b) antibody infilling models that additionally use structural information. This would be a very unexpected result that shows that the task of antibody sequence infilling is actually surprisingly trivial and can be solved with minimal knowledge of the problem. However, the paper fails to discuss this adequately and just presents the results as they are without further discussion of these implications.

The authors base their claim on a) good sequence recovery performance and b) highly diverse samples. While recovery is easily interpretable as it is the task that is optimized for, diversity as such has no inherent meaning, as long as it cannot be validated that the diverse samples are indeed biologically plausible. ProGen perplexity as it is presented is not conclusive of the biological plausibility, as it is not clear what perplexity behaviour can be expected for CDR loops which are highly variable and comparably unstructured.

EnglishBert should be understood as an almost-random baseline, as it has no knowledge about antibody structure aside from what can be incorporated from learning new amino acid embeddings in the presented setup. Yet, on CDR3, according to the presented metrics, it performs better than Refine-GNN. Why is that?

In the reprogramming framework, two linear layers with a large number of parameters are introduced to map from the task domain to the pretrained domain. For the baseline methods, this is not the case. How is ProtBERT considered to be a fair baseline method for the task, if it was never optimized for it and has no learnable parameters?

---

> ### Author Response · Authors · 2022-11-14
> **Response**
>
> Thank you for your review and comments.
>
> 1. Strong performance of BERT
>
> BERT models are usually trained with the masked language model (MLM) objective, where, given an incomplete input sentence, the objective is to fill in the missing parts, for which they achieved very successful results. Protein sequence infilling (specifically, CDR infilling) is exactly the same task, where a small part of the sequence is masked and the objective is to infill the missing region. For example, for SabDab dataset, the protein is on average of length 300 with CDR region is on average of length 10-15 amino acids, which is 3-5% of the protein sequence length. Note that for language modeling the masking is usually applied to 15% of the words, a much large part of the sentence. Therefore, for a well designed and trained BERT model the objective of CDR infilling is a reasonable task to address, with the high chances of success.
>
> 2. *"This would be a very unexpected result that shows that the task of antibody sequence infilling is actually surprisingly trivial and can be solved with minimal knowledge of the problem".*
>
> We disagree with this statement. It is not clear what are the real biological functionality and the feasibility of the proteins with the generated CDR sequences (have to be verified in wet-lab experiments). The proposed method merely addresses the problem from the statistical point of view, looking for patterns in training data and transferring them to the test sequences.
>
> 3. *“EnglishBert should be understood as an almost-random baseline”.*
>
> We want to emphasize that EnglishBert is fine-tuned on the downstream application task of CDR inifilling, where ALL of its parameters, including the new initialized protein embeddings are trainable.  Therefore, it is improper to call this a random baseline. In fact, if we do take the original EnglishBert and replace its word embeddings with random protein ones, and do no fine-tuning, we get on CDR-H3 banchmark the accuracy (AAR) of only 4.8 - this is indeed a random baseline.
>
> 4. *"How is ProtBERT considered to be a fair baseline method for the task, if it was never optimized for it and has no learnable parameters?"*
>
> ProtBert has all its parameters as learnable and during fine-tuning all these parameters are optimized (please refer to caption in Figure 5).

---

> > ### Comment · Reviewer_qyah · 2022-11-17
> > **Not sufficient**
> >
> > The authors have not yet answered all the questions listed in my original review.
> >
> > Regarding comments:
> > 1)
> > It is obvious that both are the "same task" in the sense that they reconstruct masked out tokens - but the domains are extremely different. The % masked out comparison does not help to explain the motivation for going from english language to CDRs, as opposed to going from protein sequences to CDRs.
> >
> > 2)
> > So it is not clear whether the proposed solution is an improvement over the baselines, given that the metrics are insufficient? This should be discussed and addressed in the manuscript. "merely addressing the problem from a statistical point of view" without discussing implications is insufficient when claiming improvements on a biological problem.
> >
> > 3 and 4)
> > It would be helpful if the full parameter fine-tuning was explained in the main text of the paper, and not in an appendix figure caption.

---

> > > ### Author Response · Authors · 2022-11-18
> > > **Response**
> > >
> > > Thank you for the comments.
> > >
> > > First let us cover few of the missed questions:
> > >
> > > - *Perplexity of CDR loops:*
> > > The perplexity computed using pre-trained ProGen model is the likelihood of the generated CDR and it reflects the well-formed composition and naturalness of the sequences. The perplexity of the original CDR sequences are in the range 5-6, similar to the values of the generated CDR’s, indicating their high quality.
> > >
> > > - *Why did we select EnglishBert for reprogramming rather than ProtBert?*
> > > Reprogramming usually applies to the cases when we use source model and the target task in DIFFERENT domains (English -> Protein), especially when the target domain has no good pre-trained models for fine-tuning. In case of ProtBert, the model is already in protein domain, so there is nothing to reprogram. Rather, we fine-tune ProtBert on the downstream task of antibody infilling. We agree considering the reverse Protein -> English reprogramming is an interesting idea, but it is out of our scope.
> > >
> > > - *Why are BERT models pre-trained on antibody sequences not considered?*
> > > The reason is that we did not find high performing, publicly available pre-trained BERT models on antibody sequences. As we showed in our work, using general pre-trained protein BERT model can be useful for the task of antibody infilling.
> > >
> > > Now we address your latest comments:
> > >
> > > - *Motivation for going from English language to CDRs, as opposed to going from protein sequences to CDRs.*
> > >  Again, the objective of our work was to show that reprogramming (when the model and the task are in DIFFERENT domains) can be done efficiently and can produce good performance results. As a baseline to ReprogBert, we also compared to ProtBert, which is the in-domain model, and showed that ReprogBert is able to maintain accurate amino acid recovery rates, while offering high diversity of the generated protein sequences.
> > >
> > > - *Not clear whether the proposed solution is an improvement over the baselines*
> > > We want to emphasize that from antibody protein design point of view, important metrics are Perplexity (lower the better, to reflect naturalness), RMSD/TM (lower/higher the better, to reflect structure/fold consistency), Diversity (higher the better, to reflect better coverage of the sequence space, i.e. designability) and Neurtralization ability (higher the better, to reflect functional consistency). In particular, see Section 3.5 where we presented Covid virus neutralization experiments, which showed that ReprogBert is able to improve the score more than any of the other baselines, including ProtBert and EnglishBert. All of these metrics reflect structural and functional robustness (i.e. resistant to mutation) from an evolutionary perspective (Wagner  A. Robustness and evolvability: a paradox resolved. Proc. Biol. Sci. 275:91–100.(2008)). Improvement over these metrics showing diverse sequences  corresponding to same (predicted) structure and function thus holds important biological implication, as shown in https://spj.sciencemag.org/journals/bdr/2022/9842315/ and https://www.pnas.org/doi/10.1073/pnas.1908723117
> > >
> > > - *It would be helpful if the full parameter fine-tuning was explained in the main text of the paper, and not in an appendix figure caption.*
> > > Thank you for this recommendation.

---

### Official Review · Reviewer_RNXK · 2022-10-22

**Confidence:** 4
**Correctness:** 3
**Technical Novelty And Significance:** 2
**Empirical Novelty And Significance:** 2
**Recommendation:** 3

**Clarity, Quality, Novelty And Reproducibility:**

Quality: Limited empirical improvements from reprogramming, when compared to ProtBert without reprogramming.

Clarity: Overall easy to read, with clear tables and figures. It would help to be more clear on the motivation for the problem setup. In particular, typically CDR1 and CDR2 are largely dependent on the V gene (also reflected in framework regions). What's the motivation for only designing the CDRs when the V gene is already known?

Originality: Limited technical novelty.

**Strength And Weaknesses:**

Strengths:
+ Reprogramming is an interesting idea and hasn't been previously applied to protein design.

Weaknesses:
+ Limited empirical improvements compared to ProtBert.
+ Limited in scope. The infilling problem for protein sequences could be of more general interest on other classes of proteins, although this paper is focused on the specific subarea of antibody CDR design.
+ Can the model capture interactions between the CDR loops?

**Summary Of The Paper:**

This paper introduced "Reprogramming for Protein Sequence Infilling", for the antibody CDR design problem. Starting from a more general protein language model, the model is "reprogrammed" for the infilling task (infilling for antibody CDR loops). The method is then empirically evaluated on antibodies in the SAbDab database.

**Summary Of The Review:**

Limited technical novelty and application scope. This approach is interesting and probably could be useful in practical application, but I'm not sure how useful it is to the broader community.

---

> ### Author Response · Authors · 2022-11-14
> **Response**
>
> 1. Limited improvement compared to ProtBert.
>
> On SabDab CDR-H3 dataset, ProtBert achieves 41.5 recovery and 14.5 diversity, while ReprogBert gets 32.6 in recovery and 67.4 in diversity. ReprogBert is able to maintain high recovery rate (>30%, a commonly used threshold), while significantly increasing CDR sequence diversity, a valuable property in antibody design.  Also, please note that CDR-H3 design is the most important and most difficult design task due to longer length and sequence variance.
>
> When designing the infilling of all three regions, ReprogBert is also able to maintain relatively high recovery while adding even more diversity to the generated sequences (57.4 vs 8.2 for ProtBert).
>
> On the task of neutralizing SARS-CoV-2 virus, ReprogBert gets 76.7 score vs the 74.7 for ProtBert.
>
> We are unsure why these significant results are called "limited improvement"?
>
> Also, please note that all three methods ProtBert, EnglishBert and ReprogBert are the three proposed approaches for CDR infilling, which were presented in our work. The closest existing  baseline is Refine-GNN. For example, on SabDab CDR-H3 task, Refine GNN achieves 28.2 (vs 32.6 for ReprogBert) in recovery and 25.7 (vs 67.4 for ReprogBert) in diversity. Again, it is not clear what is "limited improvement" in this case?
>
> 2. Limited in scope.
>
> Indeed the method is general and can be applied to protein sequence infilling, however we focused on the special task of CDR design as a specific subtask with much attention in recent years.

---

### Official Review · Reviewer_UBBq · 2022-10-29

**Confidence:** 5
**Clarity, Quality, Novelty And Reproducibility:** novelty is not enough.
**Correctness:** 2
**Technical Novelty And Significance:** 2
**Empirical Novelty And Significance:** 2
**Recommendation:** 3

**Strength And Weaknesses:**

Strength:
1. The writing is clear and the authors give a clear motivation for the work.
2. The results show improvements in the work.

Weakness:
1. Novelty is really limited. The authors do not give enough modifications to the model, while it is interesting to use the English BERT model, but the method seems lack enough motivation when comparing the MR method with protein pre-trained model.
2. Similar to the above one, the results are interesting that the MR is better than protein pre-trained model, but the reason is not explained and the study is not enough. This strong lacks enough content for this paper.
3. More studies and results are expected for the later version.
4. Code is not released, though the results are promising.

**Summary Of The Paper:**

This paper proposes a framework for the antibody sequence design task. The main idea is to use the pre-trained model for sequence prediction. The difference is that they use a pre-trained English BERT model with some modifications. The results show improvements of the sequence prediction.

Overall speaking, this is an easy work.


**Summary Of The Review:**

See above.

---

> ### Author Response · Authors · 2022-11-14
> **Response**
>
> The reviewer does state that "the results show improvements in the work", but complains about novelty. The novelty is in reprogramming pretrained language models to the task of antibody sequence infilling via lightweight cross-domain training. Was there such an approach available for the task of CDR infilling before? No. Is the method effective, and achieve promising results outperforming baselines in terms of recovery and diversity rates of CDR? Yes. We hope the reviewer can take a fresh look of our submission and re-evaluate the contributions of this work.
>
> Please note that this review directly contradicts R4, which states that "It is a novel and original approach that has seen no prior application in biological sequence design".

---

> > ### Comment · Reviewer_UBBq · 2022-11-17
> > **Not satisfied rebuttal**
> >
> > I am not satisfied with what the author responded. Please noted that the term reprogramming is indeed what mask pre-training in BERT.
> > At last, I didn't see many differences between BERT infilling and reprogramming. The results are improved, but the novelty is limited. These two aspects are not related.
> > More importantly, the authors also do not provide any more studies.
> > I agree with other reviewers that the novelty is still limited.

---

> > > ### Author Response · Authors · 2022-11-18
> > > **Response**
> > >
> > > Thank you for your comment.
> > > The term ***reprogramming*** is used in the literature in a sense which is quite different from ***fine-tuning*** (See Elsayed et. al, Adversarial reprogramming of neural networks). In fine-tuning, the modality/domain of the pretrained model (for example: English/Language) and the fine-tuned model (also English/Language) is the same. In reprogramming, the pretrained model is used in an entirely different domain (the input vocabularies of the pretrained and fine-tuned domains need not match).  The main difference between our proposed fine-tuned EnglishBert and reprogrammed ReprogBert is that EnglishBert has ALL of its parameters optimized on the infilling task, while in ReprogBert only the linear projection matrices (used to map the input and output vocabularies of the different domains) are updated and the rest of the model remains fixed, thus leveraging existing knowledge of the out-of-domain pre-trained model. We think this work is novel because we could not find any other work that infills protein sequences with high accuracy and diversity using reprogramming. We would be happy to update the paper if the reviewer is able to provide us with any references.

---

> > > > ### Comment · Reviewer_UBBq · 2022-11-22
> > > > **Domain Transfer**
> > > >
> > > > I agree that the domains are totally different. But please note that this kind of learning can be regarded as domain transfer/adaption, only optimizing the newly added matrices are not indeed novel, or at least straightforward. For example, when ALBERT is doing a slight pre-training model, they compress the matrix with a new matrix (though different, but this modification in more paper is easy).
> > > > As I said, the motivation why doing such kind of transfer learning is not clear. Why learning a protein/antibody specific pre-trained model is worse than English BERT? More studies or explanations are indeed required.

---

> > > > > ### Author Response · Authors · 2022-11-23
> > > > > **Response**
> > > > >
> > > > > Thanks again for your comments.
> > > > >
> > > > > - Only optimizing the newly added matrices are not indeed novel, or at least straightforward.
> > > > >
> > > > > Optimizing projection matrices is indeed not novel (reprogramming literature uses same technique). The novelty is in applying reprogramming to the domain of antibody CDR infilling and showing its good performance in sequence recovery, diversity and protein structure consistency.
> > > > >
> > > > > - Why learning a protein/antibody specific pre-trained model is worse than English BERT?
> > > > >
> > > > > Pre-trained protein model, ProtBert, is in fact performing better than EnglishBert in terms of AAR, which is expected since ProtBert is  an in-domain model and is knowledgable about the protein sequences. So if the goal is to  solely achieve high AAR, then ProtBert is the model to go. However, as emphasized in the paper, the goal of CDR design, and protein design in general, is to design sequences that are diverse and maintain the native structure.  Our experiments show that fine-tuning just the two linear projection matrices already enables fairly high AAR (> 30%), while also generating structure-consistent protein sequences with the highest diversity rate. Moreover, tuning just two linear projection matrices in ReprogBert has a benefit in low-data regimes, making ReprogBert very data efficient, as compared to other baselines.
> > > > >
> > > > > Finally, we note that in terms of sequence diversity, the reprogramming of BERT model with two linear projection matrices preserves better the original functionality of the  pre-trained model, as compared to fine-tuning. In a sense, standard fine-tuning learns to  map to a task-specific constrained space more accurately, leading to higher AAR but lower DIV, whereas reprogramming learns to map some  (linguistic)  features across domain, allowing more diverse but consistent generation.

---

### Official Review · Reviewer_9eKN · 2022-11-03

**Confidence:** 2
**Correctness:** 2
**Technical Novelty And Significance:** 1
**Empirical Novelty And Significance:** 2
**Recommendation:** 3

**Clarity, Quality, Novelty And Reproducibility:**

Novelty

For me, it’s difficult to gauge the novelty since the authors completely missed the “find tuning” literature. From my perspective so far, there is no novelty.

Quality

The technical work lacks context, as already mentioned.

Clarity

Aside from the lack of context, the work is clear.

Reproducibility

I didn't seen any code or data, so I guess this isn't very reproducible.



**Strength And Weaknesses:**

To me, the main weakness of this work is the lack of reference to the “fine tuning” literature.

**Summary Of The Paper:**

In this work, the authors reappropriate the term “fine tuning” and call it “reprogramming”. The main aim of the work is to propose new CDR loops as an infilling approach Some experimental results suggest that the proposed approach outperforms the state of the art.


**Summary Of The Review:**

This paper misses a lot of relevant context, so it’s difficult to put in context. I would recommend the authors find relevant “fine tuning” papers and start from there.

---

> ### Author Response · Authors · 2022-11-14
> **Response**
>
> 1. Lack of novelty and fine-tuning literature.
>
> The objective of our work is to adopt a large pretained language model in English language domain to the domain of antibody sequence infilling (CDR). We fine-tune an English LLM via model reprogramming to antibody language, and compare the performance with (1) English LLM fine-tuned on antibody language (2) Protein LLM fine-tuned on antibody language on CDR infilling task. We reviewed the current literature on protein design, antibody CDR design, reprogramming to position our work. It is not clear what the reviewer refers by "completely missed the fine tuning literature". Please clarify. If there is work that we missed, we would appreciate if they point us to it. Please note that the literature on reprogramming or fine-tuning language models is vast but only few of them are relevant to the current task of protein sequence infilling, therefore we covered it selectively.
>
> Please note that the claim that there is no novelty directly contradicts R4, which states that "It is a novel and original approach that has seen no prior application in biological sequence design".
>
> 2. Reproducibility.
>
> The reviewer states that "I didn't seen any code or data, so I guess this isn't very reproducible". We disagree with this statement. Moreover, in Appendix, Section C we tried to provide many relevant parts on model training and architecture.

---

### Decision · Program_Chairs · 2023-01-20

**Decision:**

Reject

**Justification For Why Not Higher Score:**

Quoting: The reviewers did not find this work sufficiently novel, or the results convincingly strong to recommend publication. The work could be strengthened by acknowledging existing finetuning literature, bolstering performance comparisons.

**Justification For Why Not Lower Score:**

N/A

**Metareview: Summary, Strengths And Weaknesses:**

This work studies the application of pretrained language models to antibody sequence prediction. Finetuning / domain adaptation of pre-trained language models is a valuable technique and it is interesting to apply this to antibody sequence prediction. The reviewers did not find this work sufficiently novel, or the results convincingly strong to recommend publication. The work could be strengthened by acknowledging existing finetuning literature, bolstering performance comparisons.